# Using Tree Height, Crown Area and Stand-Level Parameters to Estimate Tree Diameter, Volume, and Biomass of *Pinus radiata*, *Eucalyptus globulus* and *Eucalyptus nitens*

Carlos A. Gonzalez-Benecke [1],*, M. Paulina Fernández [2,3,4], Jorge Gayoso [5], Matias Pincheira [6] and Maxwell G. Wightman [1,7]

1. Department of Forest Engineering, Resources and Management, College of Forestry, Oregon State University, Corvallis, OR 97331, USA
2. Departamento de Ecosistemas y Medio Ambiente, Facultad de Agronomía e Ingeniería Forestal, Pontificia Universidad Católica de Chile, Santiago 7820436, Chile
3. Centro Nacional de Excelencia para la Industria de la Madera (CENAMAD), Pontificia Universidad Católica de Chile, Santiago 7820436, Chile
4. Centro UC de Innovación en Madera, Pontificia Universidad Católica de Chile, Santiago 7820436, Chile
5. Facultad de Ciencias Forestales y Recursos Naturales, Universidad Austral de Chile, Valdivia 5110566, Chile
6. Forestal Mininco SpA, CMPC, Los Ángeles 4440000, Chile
7. Washington State Department of Natural Resources, Olympia, WA 98504, USA
* Correspondence: carlos.gonzalez@oregonstate.edu; Tel.: +541-737-2103; Fax: +541-737-1277

**Abstract:** Accurate estimates of tree diameter, height, volume, and biomass are important for numerous economic and ecological applications. In this study, we report exponential equations to predict tree DBH (cm), stem volume over bark (VOB, m$^3$), and total above-stump biomass (TASB, kg) using three varying levels of input data for *Pinus radiata* D. Don, *Eucalyptus globulus* Labill., and *Eucalyptus nitens* (H.Deane & Maiden) Maiden planted trees. The three sets of input data included: (1) tree height (HT, m), (2) tree HT and ground projected living crown area (CA, m$^2$), and (3) tree HT, CA, and additional stand parameters. The analysis was performed using a large dataset covering the range of distribution of the species in central Chile and included stands of varying ages and planting densities. The first set of equations using only HT were satisfactory with Adj-R$^2$ values ranging from 0.78 to 0.98 across all species and variables. For all three species, estimation of DBH, VOB, and TASB as a function of HT improved when CA was added as an additional independent variable, increasing Adj-R$^2$ and reducing RMSE. The inclusion of stand variables, such as age and stand density, also resulted in further improvement in model performance. The models reported in this study are a robust alternative for DBH, VOB, and TASB estimations on planted stands across a wide range of ages and densities, when height and CA are known, especially when input data are derived from remote sensing techniques.

**Keywords:** radiata pine; blue gum; shinning gum; diameter-height allometry; crown diameter; stem volume; above-ground biomass; growth and yield modeling; remote sensing

## 1. Introduction

Accurate estimates of tree diameter, height, volume, and biomass are important for numerous economic and ecological applications, such as describing stand structure, determining the merchantability of stands, and calculating the carbon stock of forests [1,2]. Historically, these metrics have been determined through a combination of direct field sampling and allometric equations. Forest inventories typically involve field-based measurements of tree height and diameter and allometric equations to predict tree biomass from these measurements [3,4]. Conducting inventories of a large number of individual trees, however, can be an expensive and a time-consuming process [5]. Tree height can often be especially difficult to measure at a large scale and, due to this, foresters often use

allometric equations to predict tree height from tree diameter at breast height (DBH) [4]. Recently, there has been a growing interest in the use of remote sensing as an alternative to field-based forest inventories due to the lower labor requirements of the former. Remote sensing also offers the advantage of providing estimates of forest characteristics at larger spatial and temporal scales [6].

Light detection and ranging (LiDAR) and aerial photogrammetry (AP) are remote sensing techniques well suited to characterizing numerous tree characteristics, such as height, crown area, crown height, leaf area, and aboveground biomass [7–12]. These techniques can provide accurate estimates of individual tree height; however, tree DBH is not directly imaged by LiDAR or AP. Therefore, in order for LiDAR or AP to estimate aboveground biomass, equations to derive tree DBH from other tree and stand variables must be used, as individual tree biomass equations typically require DBH [11]. Research has shown that tree DBH is well correlated with crown radius and stem height [13–18]. It is also likely that the inclusion of stand-level variables could improve DBH prediction equations, as crown, stem, and stand attributes are interrelated [16,19].With the proper algorithms, LiDAR or AP can also provide estimates of stand density using individual tree detection and crown delineation [20,21].

Equations to predict tree DBH from tree height and stand parameters has been reported for several important tree species [5,11,22–24]; however, there is currently limited information for *Eucalyptus globulus* Labill. (*E. globulus*), *Eucalyptus nitens* (H.Deane & Maiden) Maiden (*E. nitens*), and *Pinus radiata* D. Don (*P. radiata*). Furthermore, the functions reported by Bi et al. [22] for *P. radiata* use only tree height to estimate DBH and do not include crown area or any stand attribute.

Traditional inventory data are often the key input used for silvicultural decision making, harvesting planning, growth projections, site quality assessments, fuel evaluations, and risks assessments, among other uses. The use of unmanned aerial vehicles (UAVs) with Lidar, RGB, and/or multispectral cameras, however, is becoming an increasingly useful tool for forest monitoring [25], allowing for larger scale, more frequent, and lower cost assessments, in comparison with traditional ground-based forest inventories. This is particularly true in regions with poor accessibility. Advancements in remote sensing now allows for the observation of structural, compositional, and functional forest attributes in near-real-time, which will enable more agile, adaptive, and efficient silviculture [26], which is important under the present circumstances of changing social and environmental conditions. Therefore, it is important to incorporate the new information acquired by these new techniques into decision support systems that have historically relied on traditional inventory data.

The species in this study represent the most important commercial tree species in Chile and many other countries due to their high yield and plasticity across a wide range of environmental conditions [27,28]. There are over two million ha of *P. radiata* and *Eucalyptus* plantations in Chile alone [29]. Developing equations to predict the DBH, volume, and biomass of these species from LiDAR (or other remote sensing techniques), the estimated height would prove valuable for estimating the merchantability of stands and determining the amount of carbon sequestration provided by plantations, especially considering the extent of these species across the world.

The main objective of this study was to produce equations to estimate DBH, stem volume, and whole-tree above-stump biomass for individual *E. globulus*, *E. nitens*, and *P. radiata* trees using tree height, crown area, and stand attributes as independent variables. The specific objectives were (1) to develop equations for the three species to predict tree DBH, stem volume, and whole-tree above-stump biomass using height alone; (2) to determine if the inclusion of crown area in these functions improved model predictions; and (3) to assess if the inclusion of stand level attributes generates further improvements in the prediction functions.

## 2. Materials and Methods

### 2.1. Data Description

The dataset used in this study was previously reported by Gonzalez-Benecke et al. [30] and consisted of two sources previously used to publish general biomass and volume functions in Chile [31,32] and unpublished data provided by the Chilean forestry company CMPC. The dataset consisted of 200 *E. globulus*, 119 *E. nitens*, and 316 *P. radiata* trees measured at 22 (*E. globulus*), 20 (*E. nitens*), and 55 (*P. radiata*) sites, respectively. The data were collected under different site, age, and management conditions, reflecting a variety of silvicultural inputs (planting density, soil preparation, fertilization, weed control, and thinning), site characteristics (physiographic region, soil type, and climate), genetics, and developmental stage. The stand characteristics at the time of sampling were thought to integrate differences in allometry due to varying silviculture practices, site qualities, and stand ages. Details on site conditions and sampling procedures can be found in each of the publications previously mentioned. Sample trees included the range of sizes encountered at each study.

The data covered the geographic range where the species are planted in central Chile (Figure 1), spanning 750 km from North to South. The climate ranged between warm-summer Mediterranean (Csb) and temperate oceanic (Cfb) climates. The range of rainfall was between 660 and 2280 mm year$^{-1}$ and mean annual temperatures ranged between 10.4 to 14 °C [33]. Trees ages ranged from 2 to 13 (*E. globulus*), 2 to 16 (*E. nitens*), and 1 to 24 (*P. radiata*) years old with diameter outside bark at 1.3 m height (DBH, cm) and total height (HT, m) ranging between 2.1 to 48.5 cm and 1.3 to 39.0 m, respectively (Table 1).

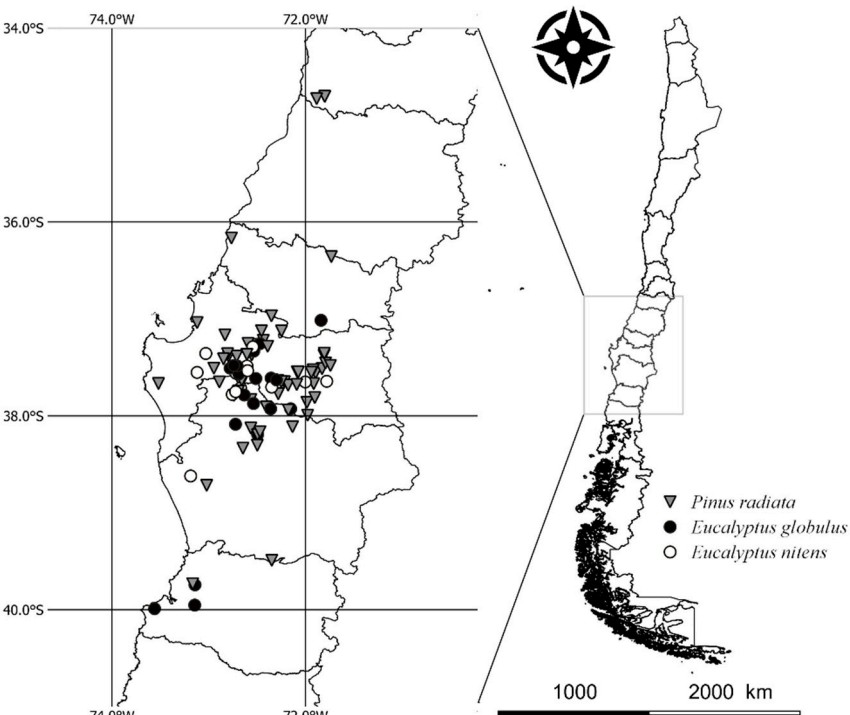

**Figure 1.** Location of sample sites for *E. globulus* (filled circle), *E. nitens* (open circle), and *P. radiata* (grey triangle) trees growing in central Chile (map includes administrative divisions of the country).

The dataset had several tree-level attributes, including DBH, HT, living crown width (CW, m), dry weight of the whole-tree above-stump biomass (TASB, kg), and total stem volume over bark (VOB, m$^3$). CW was measured in two opposite directions and ground projected living crown area (CA, m$^2$) was calculated based on CW measurements assuming the shape as an ellipse. Further details on TASB determination can be found in Gonzalez-Benecke et al. [30]. In addition to tree-level attributes, the dataset also contained several

stand-level attributes, including the number of living trees per hectare (TPH, ha$^{-1}$) and stand age (AGE, years) (Table 1). TPH was not obtained for 3 stands of *E. globulus*, 4 stands of *E. nitens*, and 6 stands of *P. radiata*. Site index (SI) was not available as a stand-level attribute. A comparison among species for the general relationships of HT and CA with DBH, TASB, and VOB is presented in Figure 2.

**Table 1.** Summary statistics of measured *E. globulus* (*n* = 200), *E. nitens* (*n* = 119), and *P. radiata* (*n* = 316, trees.

| Species | Attribute | Unit | Mean | StdDev | Minimum | Maximum |
|---|---|---|---|---|---|---|
| *E. globulus* | AGE | year | 7.0 | 3.7 | 2.3 | 13.3 |
| | HT | m | 14.8 | 7.4 | 3.8 | 32.8 |
| | DBH | cm | 13.2 | 5.4 | 3.4 | 27.0 |
| | CW | m | 2.8 | 1.0 | 0.7 | 5.4 |
| | CA | m$^2$ | 6.8 | 4.7 | 0.4 | 22.8 |
| | BA | m$^2$ ha$^{-1}$ | 17.1 | 10.6 | 0.6 | 44.4 |
| | TPH | trees ha$^{-1}$ | 1269 | 292 | 625 | 1960 |
| | TASB | kg | 104.5 | 148.3 | 1.7 | 1100.9 |
| | VOB | m$^3$ | 0.152 | 0.227 | 0.002 | 1.680 |
| *E. nitens* | AGE | year | 8.0 | 4.4 | 2.3 | 16.3 |
| | HT | m | 19.9 | 8.1 | 6.2 | 38.8 |
| | DBH | cm | 17.3 | 7.0 | 5.5 | 34.5 |
| | CW | m | 3.8 | 1.0 | 1.9 | 7.3 |
| | CA | m$^2$ | 12.5 | 7.0 | 2.8 | 42.2 |
| | BA | m$^2$ ha$^{-1}$ | 29.3 | 18.0 | 3.9 | 62.7 |
| | TPH | trees ha$^{-1}$ | 1139 | 116 | 919 | 1408 |
| | TASB | kg | 180.1 | 187.9 | 10.8 | 823.2 |
| | VOB | m$^3$ | 0.25 | 0.28 | 0.01 | 1.15 |
| *P. radiata* | AGE | year | 14.0 | 6.5 | 1.3 | 24.3 |
| | HT | m | 19.8 | 10.1 | 1.3 | 38.0 |
| | DBH | cm | 23.0 | 10.9 | 2.1 | 48.5 |
| | CW | m | 3.6 | 1.4 | 0.5 | 8.3 |
| | CA | m$^2$ | 11.9 | 9.0 | 0.2 | 54.1 |
| | BA | m$^2$ ha$^{-1}$ | 27.8 | 17.4 | 0.7 | 65.9 |
| | TPH | trees ha$^{-1}$ | 739 | 323 | 220 | 1600 |
| | TASB | kg | 238.2 | 230.6 | 0.4 | 972.8 |
| | VOB | m$^3$ | 0.52 | 0.52 | 0.00 | 2.39 |

AGE: tree age (yrs.); HT: total tree height (m); DBH: outside bark diameter at 1.3 m height (cm); CW: living crown width (m); CA: ground projected living crown area (m$^2$); BA: basal area (m$^2$ ha$^{-1}$); TPH: number of living trees per hectare (ha$^{-1}$); TASB: total above-stump biomass (kg); VOB: stem volume over bark (m$^3$).

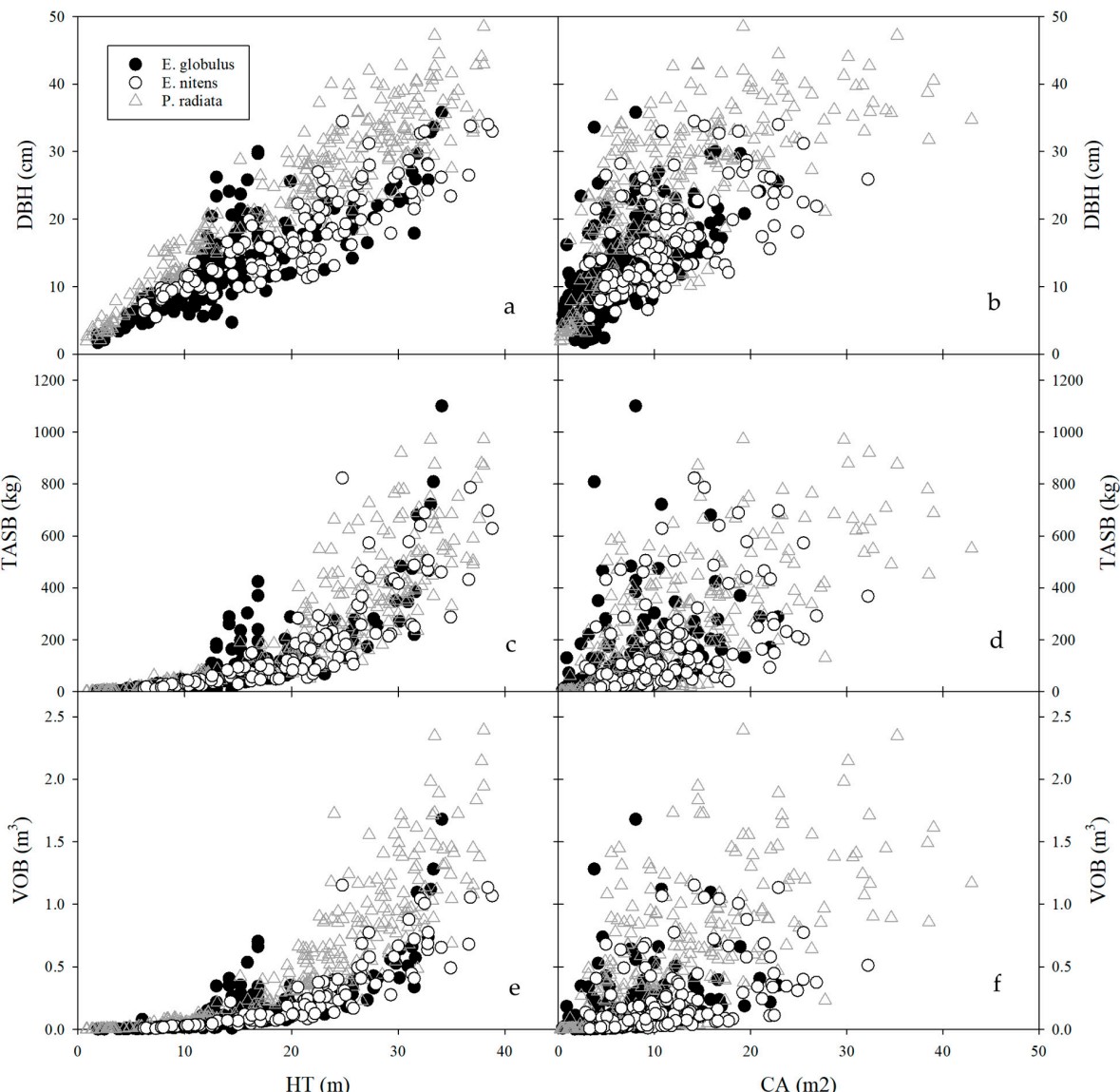

**Figure 2.** Relationships between total tree height (HT, m) and ground projected crown area (CA, m$^2$) with stem outside bark diameter at 1.3 m height (DBH, cm) (**a**,**b**), total above-stump biomass (TASB, kg) (**c**,**d**), and stem volume over bark (VOB, m$^3$) (**e**,**f**) for *E. globulus* (filled circle), *E. nitens* (open circle), and *P. radiata* (grey open triangle) trees growing in central Chile.

### 2.2. Model Description

We defined three sets of equations to estimate DBH, VOB, and TASB that depended on data availability.

Set I. When only HT is known:

The model to estimate DBH, VOB, and TASB when only HT is known was:

$$DBH = a_1 \cdot (HT - 1.3)^{a_2} + \varepsilon_i \tag{1}$$

$$TASB, \ VOB = a_1 \cdot (HT^{a_2}) + \varepsilon_i \tag{2}$$

where $a_1$ and $a_2$ are curve fit parameter estimates, and $\varepsilon_i$ is the error term, with $\varepsilon_i \sim N(0, \sigma_i^2)$.

Set II: When HT and CA are known:

The model to estimate DBH, VOB, and TASB when HT and CA are known was:

$$DBH = b_1 \cdot (HT - 1.3)^{b_2} \cdot \left(CA^{b_3}\right) + \varepsilon_i \tag{3}$$

$$\text{TASB, VOB} = b_1 \cdot \left(\text{HT}^{\,b_2}\right) \cdot \left(\text{CA}^{b_3}\right) + \varepsilon_i \tag{4}$$

where $b_1$ to $b_3$ are curve fit parameter estimates, and $\varepsilon_i$ is the error term, with $\varepsilon_i \sim N(0, \sigma_i^2)$.

Set III: When HT, CA, AGE, and TPH are known:

Model fitting was conducted for all equations that include AGE and/or TPH in addition to HT and CA. Within each species and for each of the DBH/biomass/volume components, the model with significant parameters estimates for AGE and/or TPH with the lowest AIC was finally selected. In order to detect multicollinearity among explanatory variables, the variance inflation factor (VIF) was monitored, discarding variables included in the model with VIF larger than 5 [34].

For *E. globulus*, the models to estimate DBH, VOB, and TASB when HT, CA, and the stand-level variables AGE and TPH are known were:

$$\text{DBH} = c_1 \cdot (\text{HT} - 1.3)^{c_2} \cdot (\text{CA}^{c_3}) \cdot (\text{AGE}^{c_4}) + \varepsilon_i \tag{5}$$

$$\text{TASB, VOB} = c_1 \cdot (\text{HT}^{\,c_2}) \cdot (\text{CA}^{c_3}) \cdot (\text{AGE}^{c_4}) \cdot (\text{TPH}^{c_5}) + \varepsilon_i \tag{6}$$

For *E. nitens*, the models to estimate DBH, VOB, and TASB when HT, CA, and the stand-level variables AGE and TPH are known were:

$$\text{DBH} = c_1 \cdot (\text{HT} - 1.3)^{c_2} \cdot (\text{CA}^{c_3}) \cdot (\text{TPH}^{c_5}) + \varepsilon_i \tag{7}$$

$$\text{TASB} = c_1 \cdot (\text{HT}^{\,c_2}) \cdot (\text{CA}^{c_3}) \cdot (\text{AGE}^{c_4}) \cdot (\text{TPH}^{c_5}) + \varepsilon_i \tag{8}$$

$$\text{VOB} = c_1 \cdot (\text{HT}^{\,c_2}) \cdot (\text{CA}^{c_3}) \cdot (\text{AGE}^{c_4}) + \varepsilon_i \tag{9}$$

For *P. radiata*, the models to estimate DBH, VOB, and TASB when HT, CA, and the stand-level variables AGE and TPH are known were:

$$\text{DBH} = c_1 \cdot (\text{HT} - 1.3)^{c_2} \cdot (\text{CA}^{c_3}) \cdot (\text{TPH}^{c_5}) + \varepsilon_i \tag{10}$$

$$\text{TASB} = c_1 \cdot (\text{HT}^{\,c_2}) \cdot (\text{CA}^{c_3}) \cdot (\text{AGE}^{c_4}) \cdot (\text{TPH}^{c_5}) + \varepsilon_i \tag{11}$$

$$\text{VOB} = c_1 \cdot (\text{HT}^{\,c_2}) \cdot (\text{CA}^{c_3}) \cdot (\text{TPH}^{c_5}) + \varepsilon_i \tag{12}$$

where $c_1$ to $c_6$ are curve fit parameter estimates, and $\varepsilon_i$ is the error term, with $\varepsilon_i \sim N(0, \sigma_i^2)$.

### 2.3. Model Fitting and Evaluation

All statistical analyses were performed using SAS 9.4 (SAS Inc., Cary, NC, USA). For all parameter estimates reported, non-linear model fitting was carried out using the procedure PROC NLIN. As the number of sampled trees per plot was seven or less trees, plot effect was not included in the analysis as we assumed that trees were taken from spatially independent locations. We used a 10-fold cross validation [34] to evaluate the predictive ability of all equations, randomly splitting the dataset into 10 subsets with approximately equal numbers of observations. In the 10-fold cross validation two measures of accuracy were used for each variable: (i) root mean square difference (RMSD) and (ii) mean bias error (Bias, predicted-observed).

## 3. Results

### 3.1. Model Fitting

The prediction equations and parameter estimates to calculate DBH, TASB, and VOB using tree HT, CA, and stand-level parameters for *E. globulus*, *E. nitens*, and *P. radiata* trees growing in central Chile are presented in Tables 2–4. All parameter estimates were significant at $p < 0.05$. The models to calculate DBH, TASB, and VOB using tree HT alone (model set I) had satisfactory predictive power for all species and variables. The models to estimate DBH from HT alone showed a better fit than for TASB and VOB with Adj-$R^2$ values ranging from 0.933 for *E. globulus* to 0.975 for *P. radiata*. The Adj-$R^2$ values for TASB

and VOB were still high for all species ranging from 0.784 (VOB for *E. globulus*) to 0.901 (VOB for *P. radiata*).

**Table 2.** Parameter estimates and fit statistics of the selected general DBH/biomass/volume functions for *E. globulus* trees growing in Central Chile.

| Model Set | Variable | Model | Parameter | Parameter Estimate | SE | Adj-$R^2$ | RMSE | CV % |
|---|---|---|---|---|---|---|---|---|
| I | DBH | $= a_1 \cdot (\text{HT} - 1.3)^{a_2}$ | $a_1$ | 2.075475 | 0.241144 | 0.933 | 3.85 | 29.1 |
| | | | $a_2$ | 0.737353 | 0.040622 | | | |
| | TASB | $= a_1 \cdot (\text{HT}^{a_2})$ | $a_1$ | 0.07582 | 0.040413 | 0.813 | 78.39 | 75.0 |
| | | | $a_2$ | 2.54799 | 0.160292 | | | |
| | VOB | $= a_1 \cdot (\text{HT}^{a_2})$ | $a_1$ | 0.000064929 | 0.000042188 | 0.784 | 0.127 | 83.5 |
| | | | $a_2$ | 2.712235 | 0.19464 | | | |
| II | DBH | $= b_1 \cdot (\text{HT} - 1.3)^{b_2} \cdot \left(\text{CA}^{b_3}\right)$ | $b_1$ | 1.766367 | 0.181048 | 0.953 | 3.24 | 24.5 |
| | | | $b_2$ | 0.642174 | 0.036413 | | | |
| | | | $b_3$ | 0.224741 | 0.026321 | | | |
| | TASB | $= b_1 \cdot \left(\text{HT}^{b_2}\right) \cdot \left(\text{CA}^{b_3}\right)$ | $b_1$ | 0.059127 | 0.029441 | 0.833 | 74.04 | 70.8 |
| | | | $b_2$ | 2.452531 | 0.14752 | | | |
| | | | $b_3$ | 0.281297 | 0.060094 | | | |
| | VOB | $= b_1 \cdot \left(\text{HT}^{b_2}\right) \cdot \left(\text{CA}^{b_3}\right)$ | $b_1$ | 0.000050198 | 0.000031331 | 0.797 | 0.123 | 80.8 |
| | | | $b_2$ | 2.642533 | 0.184052 | | | |
| | | | $b_3$ | 0.24399 | 0.067749 | | | |
| III | DBH | | $c_1$ | 1.909369 | 0.195751 | 0.956 | 3.12 | 23.6 |
| | | $= c_1 \cdot (\text{HT} - 1.3)^{c_2} \cdot (\text{CA}^{c_3}) \cdot (\text{AGE}^{c_4})$ | $c_2$ | 0.380361 | 0.07423 | | | |
| | | | $c_3$ | 0.213189 | 0.024686 | | | |
| | | | $c_4$ | 0.307518 | 0.078219 | | | |
| | TASB | | $c_1$ | 1.60693 | 1.678005 | 0.862 | 67.22 | 64.3 |
| | | $= c_1 \cdot (\text{HT}^{c_2}) \cdot (\text{CA}^{c_3}) \cdot (\text{AGE}^{c_4}) \cdot (\text{TPH}^{c_5})$ | $c_2$ | 1.522111 | 0.188442 | | | |
| | | | $c_3$ | 0.353128 | 0.061234 | | | |
| | | | $c_4$ | 0.42111 | 0.22933 | | | |
| | | | $c_5$ | −0.233854 | 0.150093 | | | |
| | VOB | | $c_1$ | 0.000044365 | 0.000023945 | 0.837 | 0.110 | 72.4 |
| | | $= c_1 \cdot (\text{HT}^{c_2}) \cdot (\text{CA}^{c_3}) \cdot (\text{AGE}^{c_4})$ | $c_2$ | 1.496324 | 0.220388 | | | |
| | | | $c_3$ | 0.248755 | 0.05662 | | | |
| | | | $c_4$ | 1.510607 | 0.215681 | | | |

HT: total tree height (m); DBH: outside bark diameter at 1.3 m height (cm); CA: grown projected living crown area ($m^2$); AGE: tree/stand age (yrs.); TPH: number of living trees per hectare ($ha^{-1}$); TASB: total above-stump biomass (kg); VOB: stem volume over bark ($m^3$); SE: standard error; Adj-$R^2$: adjusted coefficient of determination; RMSE: root mean square error (same units as variable); CV: coefficient of variation of the RMSE ($100 \cdot \text{RMSE}/\text{mean}$). For all parameter estimates, $p < 0.05$.

**Table 3.** Parameter estimates and fit statistics of the selected general DBH/biomass/volume functions for *E. nitens* trees growing in Central Chile.

| Model Set | Variable | Model | Parameter | Parameter Estimate | SE | Adj-$R^2$ | RMSE | CV % |
|---|---|---|---|---|---|---|---|---|
| I | DBH | $= a_1 \cdot (\text{HT} - 1.3)^{a_2}$ | $a_1$ | 1.593724 | 0.246016 | 0.964 | 3.57 | 20.6 |
| | | | $a_2$ | 0.819012 | 0.049207 | | | |
| | TASB | $= a_1 \cdot (\text{HT}^{a_2})$ | $a_1$ | 0.108287 | 0.062418 | 0.864 | 95.82 | 53.2 |
| | | | $a_2$ | 2.398404 | 0.168986 | | | |
| | VOB | $= a_1 \cdot (\text{HT}^{a_2})$ | $a_1$ | 0.000061349 | 0.00003386 | 0.881 | 0.130 | 51.9 |
| | | | $a_2$ | 2.679173 | 0.160883 | | | |
| II | DBH | | $b_1$ | 1.10718 | 0.159378 | 0.973 | 3.09 | 17.8 |
| | | $= b_1 \cdot (\text{HT} - 1.3)^{b_2} \cdot \left(\text{CA}^{b_3}\right)$ | $b_2$ | 0.771231 | 0.043692 | | | |
| | | | $b_3$ | 0.20542 | 0.029766 | | | |
| | TASB | $= b_1 \cdot \left(\text{HT}^{b_2}\right) \cdot \left(\text{CA}^{b_3}\right)$ | $b_1$ | 0.039541 | 0.020778 | 0.898 | 83.11 | 46.2 |
| | | | $b_2$ | 2.399547 | 0.146035 | | | |
| | | | $b_3$ | 0.396369 | 0.064146 | | | |
| | VOB | $= b_1 \cdot \left(\text{HT}^{b_2}\right) \cdot \left(\text{CA}^{b_3}\right)$ | $b_1$ | 0.000021383 | 0.000010341 | 0.915 | 0.110 | 43.8 |
| | | | $b_2$ | 2.695027 | 0.133308 | | | |
| | | | $b_3$ | 0.394597 | 0.057055 | | | |

**Table 3.** *Cont.*

| Model Set | Variable | Model | Parameter | Parameter Estimate | SE | Adj-$R^2$ | RMSE | CV % |
|---|---|---|---|---|---|---|---|---|
| III | DBH | $= c_1 \cdot (\text{HT} - 1.3)^{c_2} \cdot (\text{CA}^{c_3}) \cdot (\text{TPH}^{c_5})$ | $c_1$ | 1.220236 | 0.892918 | 0.973 | 3.04 | 17.6 |
|  |  |  | $c_2$ | 0.770766 | 0.043979 |  |  |  |
|  |  |  | $c_3$ | 0.205358 | 0.029863 |  |  |  |
|  |  |  | $c_5$ | −0.013596 | 0.10031 |  |  |  |
|  | TASB | $= c_1 \cdot (\text{HT}^{c_2}) \cdot (\text{CA}^{c_3}) \cdot (\text{AGE}^{c_4})$ | $c_1$ | 0.034478 | 0.018895 | 0.915 | 76.61 | 42.5 |
|  |  |  | $c_2$ | 1.9837 | 0.177463 |  |  |  |
|  |  |  | $c_3$ | 0.416925 | 0.05913 |  |  |  |
|  |  |  | $c_4$ | 0.600451 | 0.173335 |  |  |  |
|  | VOB | $= c_1 \cdot (\text{HT}^{c_2}) \cdot (\text{CA}^{c_3}) \cdot (\text{AGE}^{c_4})$ | $c_1$ | 0.000021001 | 0.000010266 | 0.920 | 0.107 | 42.7 |
|  |  |  | $c_2$ | 2.360592 | 0.167086 |  |  |  |
|  |  |  | $c_3$ | 0.41502 | 0.054414 |  |  |  |
|  |  |  | $c_4$ | 0.443079 | 0.149888 |  |  |  |

HT: total tree height (m); DBH: outside bark diameter at 1.3 m height (cm); CA: grown projected living crown area (m$^2$); AGE: tree/stand age (yrs); TPH: number of living trees per hectare (ha$^{-1}$); TASB: total above-stump biomass (kg); VOB: stem volume over bark (m$^3$); SE: standard error; Adj-$R^2$: adjusted coefficient of determination; RMSE: root mean square error (same units as variable); CV: coefficient of variation of the RMSE (100·RMSE/mean). For all parameter estimates, $p < 0.05$.

**Table 4.** Parameter estimates and fit statistics of the selected local and general DBH/biomass/volume functions for *P. radiata* trees growing in central Chile.

| Model Set | Variable | Model | Parameter | Parameter Estimate | SE | Adj-$R^2$ | RMSE | CV % |
|---|---|---|---|---|---|---|---|---|
| I | DBH | $= a_1 \cdot (\text{HT} - 1.3)^{a_2}$ | $a_1$ | 2.780438 | 0.212548 | 0.975 | 4.01 | 17.4 |
|  |  |  | $a_2$ | 0.736776 | 0.023889 |  |  |  |
|  | TASB | $= a_1 \cdot (\text{HT}^{a_2})$ | $a_1$ | 0.256462 | 0.09421 | 0.890 | 109.90 | 46.1 |
|  |  |  | $a_2$ | 2.194259 | 0.107627 |  |  |  |
|  | VOB | $= a_1 \cdot (\text{HT}^{a_2})$ | $a_1$ | 0.000316672 | 0.000115223 | 0.901 | 0.232 | 44.4 |
|  |  |  | $a_2$ | 2.368601 | 0.10628 |  |  |  |
| II | DBH | $= b_1 \cdot (\text{HT} - 1.3)^{b_2} \cdot \left( \text{CA}^{b_3} \right)$ | $b_1$ | 2.393191 | 0.142812 | 0.986 | 3.05 | 13.3 |
|  |  |  | $b_2$ | 0.640371 | 0.019586 |  |  |  |
|  |  |  | $b_3$ | 0.179848 | 0.01211 |  |  |  |
|  | TASB | $= b_1 \cdot \left( \text{HT}^{b_2} \right) \cdot \left( \text{CA}^{b_3} \right)$ | $b_1$ | 0.180918 | 0.046562 | 0.947 | 76.56 | 32.1 |
|  |  |  | $b_2$ | 1.947392 | 0.076617 |  |  |  |
|  |  |  | $b_3$ | 0.445083 | 0.025744 |  |  |  |
|  | VOB | $= b_1 \cdot \left( \text{HT}^{b_2} \right) \cdot \left( \text{CA}^{b_3} \right)$ | $b_1$ | 0.000213594 | 0.000058107 | 0.947 | 0.170 | 32.6 |
|  |  |  | $b_2$ | 2.179969 | 0.080019 |  |  |  |
|  |  |  | $b_3$ | 0.389441 | 0.024942 |  |  |  |
| III | DBH | $= c_1 \cdot (\text{HT} - 1.3)^{c_2} \cdot (\text{CA}^{c_3}) \cdot (\text{TPH}^{c_5})$ | $c_1$ | 3.595495 | 0.638804 | 0.986 | 3.06 | 13.3 |
|  |  |  | $c_2$ | 0.628486 | 0.022015 |  |  |  |
|  |  |  | $c_3$ | 0.15617 | 0.014439 |  |  |  |
|  |  |  | $c_5$ | −0.048804 | 0.020124 |  |  |  |
|  | TASB | $= c_1 \cdot (\text{HT}^{c_2}) \cdot (\text{CA}^{c_3}) \cdot (\text{AGE}^{c_4}) \cdot (\text{TPH}^{c_5})$ | $c_1$ | 1.229341 | 0.549664 | 0.952 | 73.38 | 30.8 |
|  |  |  | $c_2$ | 1.702726 | 0.106506 |  |  |  |
|  |  |  | $c_3$ | 0.350139 | 0.02972 |  |  |  |
|  |  |  | $c_4$ | 0.164428 | 0.106054 |  |  |  |
|  |  |  | $c_5$ | −0.214903 | 0.041324 |  |  |  |
|  | VOB | $= c_1 \cdot (\text{HT}^{c_2}) \cdot (\text{CA}^{c_3}) \cdot (\text{TPH}^{c_5})$ | $c_1$ | 0.001177 | 0.000518775 | 0.949 | 0.168 | 32.2 |
|  |  |  | $c_2$ | 2.08249 | 0.081596 |  |  |  |
|  |  |  | $c_3$ | 0.300946 | 0.030124 |  |  |  |
|  |  |  | $c_5$ | −0.183982 | 0.039884 |  |  |  |

HT: total tree height (m); DBH: outside bark diameter at 1.3 m height (cm); CA: grown projected living crown area (m$^2$); AGE: tree/stand age (yrs); TPH: number of living trees per hectare (ha$^{-1}$); TASB: total above-stump biomass (kg); VOB: stem volume over bark (m$^3$); SE: standard error; Adj-$R^2$: adjusted coefficient of determination; RMSE: root mean square error (same units as variable); CV: coefficient of variation of the RMSE (100·RMSE/mean). For all parameter estimates, $p < 0.05$.

For *E. globulus*, the models to estimate DBH, TASB, and VOB that use only HT (model set I) were improved when CA was included (model set II) (Table 2). For example, when CA was included for DBH, the Adj-$R^2$ increased from 0.933 to 0.953, the RMSE decreased from 3.85 to 3.24 cm, and the coefficient of variation of the RMSE (CV) decreased from 29%

to 24%. For TASB and VOB, the Adj-$R^2$ increased from 0.813 to 0.833 and from 0.784 to 0.797 when CA was included, respectively. The RMSE also decreased from 78.4 to 74.0 kg, and from 0.127 to 0.123 m$^3$, for TASB and VOB, respectively. When the stand attributes AGE, and/or TPH were included (model set III), further improvements were observed. For example, for DBH, the Adj-$R^2$ increased to 0.956 and RMSE decreased to 3.12 cm. A similar trend was observed for TASB and VOB such that the Adj-$R^2$ increased to 0.863 and 0.837, and RMSE decreased to 67.2 kg and 0.110 m$^3$, respectively (Table 2). The parameter estimate for AGE was significant for all three traits, being always positive. The parameter estimate for TPH was only significant for TASB, being negative.

The models for estimating the DBH, TASB, and VOB of *E. nitens* were also improved when CA was included in addition to HT (model set II). When CA was included in the model to estimate TASB, the Adj-$R^2$ increased from 0.864 to 0.898, the RMSE decreased from 95.8 to 83.1 kg, and the CV decreased from 53 to 46% (Table 3). For DBH and VOB, the Adj-$R^2$ increased from 0.964 to 0.973 and from 0.881 to 0.915 while the RMSE decreased from 3.6 to 3.1 cm and from 0.130 to 0.110 m$^3$, respectively. Similar to *E. globulus*, when the stand attributes AGE and/or TPH were also included (model set III), further improvements were observed. For example, for TASB, the Adj-$R^2$ increased to 0.915, the RMSE decreased to 76.6 kg, and the CV decreased to 42.5%. A similar trend was observed for VOB with the Adj-$R^2$ increasing to 0.920 while the RMSE decreased to 0.107 m$^3$ (Table 3). For DBH, even though the parameter estimate for TPH was significant, it produced little improvement in model performance. When significant, the parameter estimate for TPH was negative. By contrast, when significant, the parameter estimate for AGE was positive.

The dataset of *P. radiata* included 116 pruned and 200 unpruned trees. Covariance test showed no effect of pruning condition on the relationship between CA and both DBH and HT ($p > 0.137$). Based on those results, we decided to fit a single function for both pruned and unpruned *P. radiata* trees. Similar to the other species, the models to estimate the DBH, TASB, and VOB of *P. radiata* that used only HT (model set I) were improved when CA was included (model set II) (Table 4). For example, when CA was included in the model to estimate VOB, the Adj-$R^2$ increased from 0.901 to 0.947, the RMSE decreased from 0.232 to 0.170 m$^3$, and the CV decreased from 44 to 33%. DBH and TASB also showed improvements, with Adj-$R^2$ increasing from 0.975 to 0.986 and from 0.890 to 0.952 while RMSE decreased from 4.01 to 3.05 cm and from 109.9 to 76.6 kg, respectively (Table 4). For TASB, further improvement was observed when the stand attributes AGE and TPH were also included (model set III), with Adj-$R^2$ increasing to 0.952 and RMSE decreased to 73.4 kg. For DBH and VOB, little improvement was observed when the stand attributes AGE and/or TPH were included. The parameter estimate for TPH was significant for all variables, being always negative. When significant, the parameter estimate for AGE was negative.

### 3.2. Model Evaluation

Figure 3 shows examples of model evaluation for the DBH/VOB/TASB estimations for all three species. Scatterplots of the observed against the predicted values for all three variables in model set I revealed significant heteroscedasticity, such that prediction errors tended to increase with tree size (upper panel in Figure 3). When CA was included in addition to HT (model set II), the heteroscedasticity was not observed (center panel in Figure 3). Further improvement was observed when AGE and/or TPH were included (model set III), such that the variability of DBH, TASB, and VOB was similar across the range of predicted values (bottom panel in Figure 3).

A summary of the 10-fold cross validation test used for all selected models is shown in Table 5. There was good agreement between observed and predicted values across all variables for all three species. For the models to estimate DBH/VOB/TASB using only HT as the explanatory variable (model set I), bias ranged between an 8.5% underestimation for VOB of *E. globulus* to a 2.4% overestimation for VOB of *E. nitens*. Estimates agreed better with observed values when CA was included (model set II). For example, for VOB, the RMSD was reduced from 75.6, 47.6, and 43.2% (model V1) to 72.6, 36.2, and 31.3% (model V2) for

*E. globulus*, *E. nitens*, and *P. radiata*, respectively. For TASB, the RMSD was reduced from 67.8, 48.3, and 44.8% (model B1) to 63.2, 39.3, and 31.0% (model B2) for *E. globulus*, *E. nitens*, and *P. radiata*, respectively. Further improvement was observed when AGE and/or TPH were added to the model that already included HT and CA (model set III). For example, for DBH, the RMSD was reduced from 23.8, 15.9, and 12.8% (model D2) to 23.2, 15.7, and 12.8% (model D3) for *E. globulus*, *E. nitens*, and *P. radiata*, respectively. Including stand variables AGE and/or TPH had little effect on model bias for all three species.

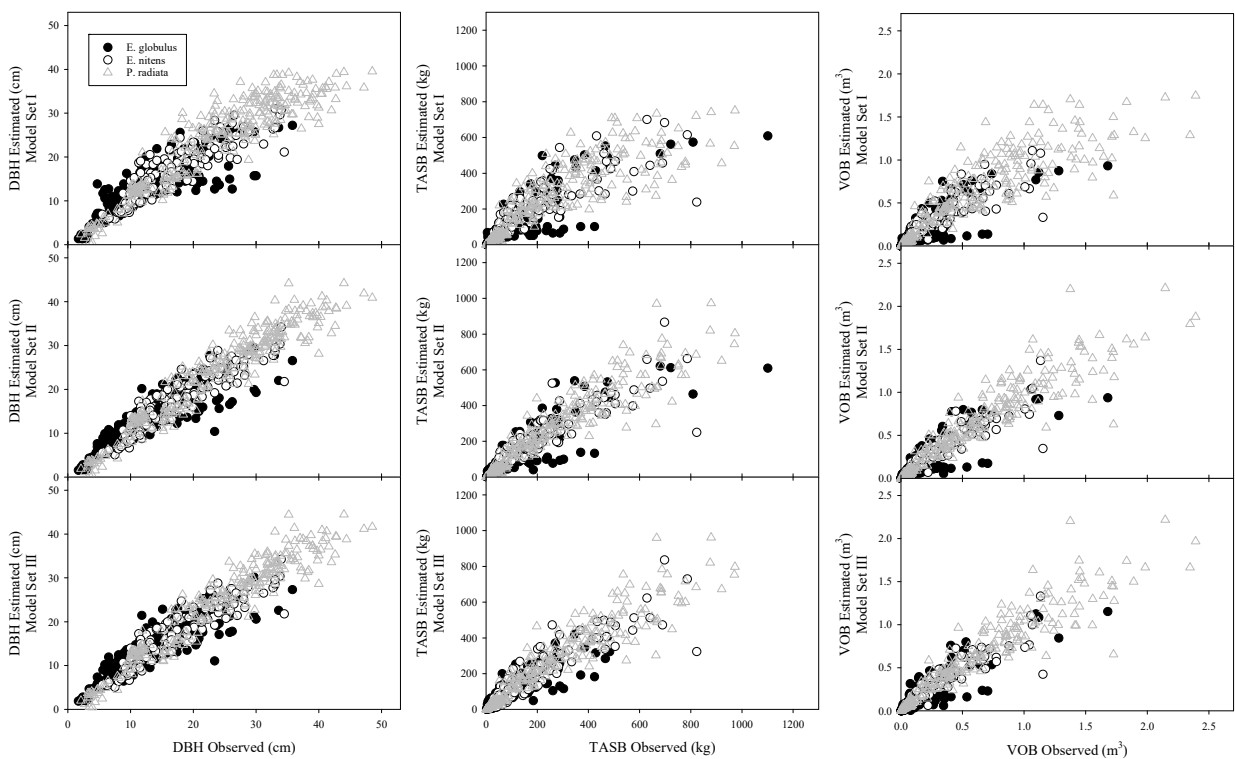

**Figure 3.** Model evaluation of DBH (**left**), TASB (**center**), and VOB (**right**) models for *E. globulus*, *E. nitens*, and *P. radiata*. Observed vs predicted (simulated) values using model set I (**upper panel**; only HT as independent variable), model set II (**center panel**; HT and CA as independent variables), and model set III (**bottom panel**; HT, CA, and AGE and/or TPH as independent variables).

**Table 5.** Summary of model evaluation statistics using 10-fold cross validation for DBH/biomass/volume estimations for *E. globulus*, *E. nitens*, and *P. radiata* trees.

| Species | Variable | Model ID | Explanatory Variables | $\bar{O}$ | $\bar{P}$ | RMSD | | Bias | |
|---|---|---|---|---|---|---|---|---|---|
| *E. globulus* | DBH | D1 | HT | 13.22 | 13.16 | 3.67 | (27.8) | 0.06 | (0.4) |
| | | D2 | HT, CA | 13.22 | 13.16 | 3.15 | (23.8) | 0.06 | (0.4) |
| | | D3 | HT, CA, AGE | 13.22 | 13.16 | 3.07 | (23.2) | 0.07 | (0.5) |
| | TASB | B1 | HT | 107.28 | 100.03 | 72.73 | (67.8) | 7.25 | (6.8) |
| | | B2 | HT, CA | 107.28 | 101.31 | 67.82 | (63.2) | 5.97 | (5.6) |
| | | B3 | HT, CA, AGE, TPH | 107.28 | 98.87 | 63.44 | (59.1) | 8.41 | (7.8) |
| | VOB | V1 | HT | 0.16 | 0.14 | 0.12 | (75.6) | 0.01 | (8.5) |
| | | V2 | HT, CA | 0.16 | 0.15 | 0.11 | (72.6) | 0.01 | (7.6) |
| | | V3 | HT, CA, AGE | 0.16 | 0.14 | 0.10 | (66.0) | 0.01 | (9.1) |
| *E. nitens* | DBH | D1 | HT | 17.23 | 17.21 | 3.37 | (19.6) | 0.03 | (0.2) |
| | | D2 | HT, CA | 17.23 | 17.13 | 2.74 | (15.9) | 0.10 | (0.6) |
| | | D3 | HT, CA, TPH | 17.23 | 17.14 | 2.70 | (15.7) | 0.09 | (0.5) |
| | TASB | B1 | HT | 177.19 | 178.68 | 85.62 | (48.3) | −1.49 | (−0.8) |
| | | B2 | HT, CA | 177.19 | 176.77 | 69.63 | (39.3) | 0.42 | (0.2) |

**Table 5.** *Cont.*

| Species | Variable | Model ID | Explanatory Variables | $\overline{O}$ | $\overline{P}$ | RMSD | | Bias | |
|---------|----------|----------|----------------------|------|------|--------|--------|-------|--------|
| | VOB | B3 | HT, CA, AGE | 177.19 | 172.34 | 68.64 | (38.7) | 4.85 | (2.7) |
| | | V1 | HT | 0.25 | 0.25 | 0.12 | (47.6) | −0.01 | (−2.4) |
| | | V2 | HT, CA | 0.25 | 0.25 | 0.09 | (36.2) | 0.00 | (−1.5) |
| | | V3 | HT, CA, AGE | 0.25 | 0.25 | 0.09 | (36.2) | 0.00 | (−0.3) |
| *P. radiata* | DBH | D1 | HT | 23.18 | 23.12 | 3.95 | (17.0) | 0.07 | (0.3) |
| | | D2 | HT, CA | 23.18 | 23.10 | 2.96 | (12.8) | 0.09 | (0.4) |
| | | D3 | HT, CA, TPH | 23.14 | 23.06 | 2.96 | (12.8) | 0.08 | (0.3) |
| | TASB | B1 | HT | 240.55 | 241.74 | 107.88 | (44.8) | −1.19 | (−0.5) |
| | | B2 | HT, CA | 240.55 | 242.12 | 74.67 | (31.0) | −1.57 | (−0.7) |
| | | B3 | HT, CA, AGE, TPH | 240.13 | 242.23 | 70.53 | (29.4) | −2.10 | (−0.9) |
| | VOB | V1 | HT | 0.53 | 0.53 | 0.23 | (43.2) | 0.00 | (−0.8) |
| | | V2 | HT, CA | 0.53 | 0.53 | 0.17 | (31.3) | 0.00 | (−0.8) |
| | | V3 | HT, CA, TPH | 0.53 | 0.53 | 0.16 | (30.5) | −0.01 | (−1.1) |

HT: total tree height (m); DBH: outside bark diameter at 1.37 m height (cm); CA: ground projected living crown area (m$^2$); AGE: tree/stand age (yrs.); TPH: number of living trees per hectare (a$^{-1}$); BA: basal area (m$^2$ ha$^{-1}$); TASB: total above-stump biomass (kg); VOB: stem volume over bark (m$^3$); $\overline{O}$: mean observed value; $\overline{P}$: mean predicted value; RMSD: root of mean square difference (same unit as observed value); Bias: mean absolute bias (predicted-observed; same unit as observed value). Values in parenthesis are percentage relative to observed mean.

## 4. Discussion

The presented functions provided adequate estimates of *E. globulus*, *E. nitens*, and *P. radiata* for DBH, TASB, and VOB using various combinations of tree height, crown area, and stand parameters, such as age and stocking. The dataset used in this analysis included a large number of individual trees with a wide range of sizes over a large geographical area in central Chile with varying soils, climates, and productivities. The dataset for *P. radiata* included pruned and unpruned trees and the reported functions were fit using both pruned and unpruned trees as we found no effect of the pruning condition on the relationship between CA and DBH and HT. The stands where tree measurements were collected also represent a wide range of ages, stocking levels, and basal areas. Due to this, we are confident that the presented functions may be applied to stands planted with these species across a broad range of sites with varying environmental and management conditions. Furthermore, the height data included in our dataset was measured directly as trees were felled for destructive sampling and, therefore, avoided potential errors associated with other methods of measuring tree height [7]. When combined with LiDAR or other remote sensing techniques capable of estimating tree height, the simple inputs, generality, and accuracy of the presented functions provide a powerful tool for assessing tree diameter, aboveground biomass, and stem volume over a large geographic area.

Three sets of functions were presented for each species and variable that varied in the number of inputs required allowing for the selection of different functions depending on desired accuracy and available inputs. The functions using only HT to predict DBH were able to explain 93%–98% of the variability in field measured DBH. This is higher than the Adj-$R^2$ of 0.77 reported by Gonzalez-Benecke et al. [5] for *P. palustris* but similar to the $R^2$ values reported by Filipescu et al. for several conifer species, which ranged between 0.84–0.91 [24]. The functions in model set I did demonstrate some degree of heteroscedasticity such that prediction errors increased with increasing tree size. This is likely due to the fact that trees tend to plateau in height as they age while continuing to grow in diameter, an effect that has been observed for several tree species [22,23]. When CA was added to the prediction function, there was no longer heteroscedasticity likely due to trees continuing to increase in crown area as they age even while HT plateaus, at least for the range of data included in this study.

The inclusion of CA increased the accuracy of the prediction functions for all the species in this study. This effect was also seen to a much greater degree in Gonzalez-

Benecke et al. [5] where the Adj-$R^2$ of functions to predict the DBH of *P. palustris* increased from 0.77 to 0.90 when CA was included in addition to HT. The Adj-$R^2$ of the functions to estimate tree DBH from HT and CA in this study (0.96–0.99) were higher than those reported elsewhere [5,11,18].

The inclusion of stand variables further improved the accuracy of the prediction functions, although to a lesser degree than the inclusion of CA. The effect of TPH and AGE on DBH was significant for all species with the exception of AGE for *E. globulus*. When significant, the parameter estimates for AGE were always positive, indicating that older trees of the same HT and CA would have larger DBH (and, hence, VOB and TASB) or higher wood density. When significant, the parameter estimates for TPH were always negative, indicating that trees of the same HT, CA, and AGE would have smaller DBH (and, hence, VOB and TASB) as stand density increases. A similar effect was reported by Pinkard and Neilsen [35] for *E. nitens*, who reported smaller TASB for trees growing in denser stands. It is possible that the inclusion of stand variables that reflect site quality, such as site index or dominant height, could further improve the functions, as these factors have been shown to be important by others [5]. Unfortunately, we were not able to include either site index or dominant height into the dataset. Nevertheless, the Adj-$R^2$ of the presented functions that included AGE/TPH was already high (0.86–0.98).

The accuracy and precision of the functions to estimate tree TASB were lower than for DBH, but still had Adj-$R^2$ values above 0.81. The functions to estimate TASB from HT alone had an average Adj-$R^2$ of 0.85, which was improved to 0.89 when CA was included. This effect was most pronounced for *P. radiata*. Our Adj-$R^2$ values are similar to the value of 0.87 reported by Popescu [11] for estimating the biomass of *P. taeda* from LiDAR derived measurements. Similar to DBH, the TASB functions were improved by including stand variables, although to a lesser degree than including CA. The effect of AGE on TASB was significant and positive for all species, reflecting that trees increase in biomass as they age. When significant (*E. globulus* and *P. radiata*), the effect of TPH on TASB was negative reflecting that, for a given HT, the biomass of individual trees of these species decreased with increasing stocking density. The lack of significance of TPH on *E. nitens* may be an effect of the reduced range of stocking on the stands of *E. nitens* sampled, where TPH varied between 919 and 1408 trees ha$^{-1}$, a smaller span when compared to *E. globulus* (625–1960 trees ha$^{-1}$) or *P. radiata* (220–1600 trees ha$^{-1}$). It is worth noting that Eucalyptus stands in Chile are managed mainly for pulp production and stands with densities outside the observed ranges are unusual [36,37].

Similar to DBH and TASB, the functions to estimate VOB were improved by adding CA and, to a lesser degree, stand variables. This effect was also seen in Gonzalez-Benecke et al. [5]. The only stand variable that was significant for *E. globulus* and *E. nitens* was AGE, which was positive and reflects the increases in stem volume seen as trees age. For *P. radiata*, TPH was significant and negative reflecting that trees of a given HT tend to have lower VOB when growing at higher densities. The wide range of stocking for *P. radiata* sampled stands, where TPH ranged between 220 and 1600 trees ha$^{-1}$, contributed to capture this effect. In Chile, *P. radiata* stands are commonly managed with one or two thinnings, with standard planting density of about 1250 to 1667 trees ha$^{-1}$, reaching a final stocking between 300 to 500 trees ha$^{-1}$ on thinned stands [37,38].

## 5. Conclusions

The reported functions have many important economic and ecological applications, especially when combined with remote sensing techniques capable of estimating tree height and projected crown area. Potential uses include estimating timber appraisal and the carbon stock of forest ecosystems, determining feedstocks for bioenergy production [39–41], evaluating fire hazard [42,43], and assessing the risk of soil erosion [40].

Even though the extensive model fitting dataset used in this study gives us confidence that the equations can be used over a wide range of sites and management conditions, we recommend using these equations within the range of the dataset shown in Table 2.

These new equations are intended as a tool to support and guide management decisions for the three species. The integration of these functions with remote sensing techniques into stand-level productivity models should enhance the flexibility and strength of the DBH, volume, and biomass predictions from those models, at a larger scale and in a more time efficient way. Future research is planned in order to use the equations for estimating stand basal area, volume, and biomass using LiDAR data.

**Author Contributions:** C.A.G.-B. and M.G.W. analyzed the data and wrote the paper. M.P.F., J.G. and M.P. leaded field measurements and wrote the paper. All authors have read and agreed to the published version of the manuscript.

**Funding:** Funding for this work was provided in part by Chilean National Commission for Scientific and Technological Research with ANID BASAL FB210015 Project Grant, and by Marie Skłodowska-Curie Research and Innovation Staff Exchange (Call:H2020-MSCARISE-2020) grant number 101007950 (Decision ES: Decision Support for the Supply of Ecosystem Services under Global Change). Funding was also provided by the Forest Engineering, Resources and Management Department at Oregon State University.

**Data Availability Statement:** Data may be available upon request to the authors.

**Acknowledgments:** The authors thank all persons who helped to collect the biomass data used in this study, including landowners and field crews. We acknowledge the support provided by Forestal Mininco SpA. CMPC Chile.

**Conflicts of Interest:** The authors declare no conflict of interest.

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
