# Peer review of "Using Tree Height, Crown Area and Stand-Level Parameters to Estimate Tree Diameter, Volume, and Biomass of Pinus radiata, Eucalyptus globulus and Eucalyptus nitens"

_forests, doi:10.3390/f13122043_

Round 1
Reviewer 1 Report
- Line 16: Please revise the typing mistake.
- Line 18-20: Can you define your equations a bit more to let the leader know what type of equation is going to present?
- Line 21: Please provide a unit for each variable.
- Line 23: Please specify which part of Chile.
- Line 24-28: Please elaborate more on the quantitative results.
- Line 67: Please provide the species of Eucalyptus.
- In terms of the objectives listed in lines 72-79, The literature review should be improved. The following studies conducted in South America could work: https://link.springer.com/article/10.1007/s11676-020-01174-y; https://link.springer.com/article/10.1007/s11676-019-01084-8
- Figure 1: Please add some toponyms. Also, the north arrow should be better presented.
- Remove the extra spaces in Equation 2.
- Line 132: Please remove "comma" after a1.
- Lines 132, 136, and 151: What is the error term?
- Line 139: than > that
- Discussion: I would remove the references to Figures and Tables in the Discussion section.
- References: Should be styled into the MDPI format.
Author Response
- Line 16: Please revise the typing mistake.
R: Fixed
- Line 18-20: Can you define your equations a bit more to let the leader know what type of equation is going to present?
R: Even though that text refers to the set of inputs and not to the structure of the equations, we indicated that the equations were exponential (line 18).
- Line 21: Please provide a unit for each variable.
R: Units added.
- Line 23: Please specify which part of Chile.
R: Central Chile added.
- Line 24-28: Please elaborate more on the quantitative results.
R: Text added in line 27.
- Line 67: Please provide the species of Eucalyptus.
R: The two species of Eucalyptus were already described in the same paragraph.
- In terms of the objectives listed in lines 72-79, The literature review should be improved. The following studies conducted in South America could work: https://link.springer.com/article/10.1007/s11676-020-01174-y; https://link.springer.com/article/10.1007/s11676-019-01084-8
R: Those papers refer are H-D relationships where DBH is known and height is to be determined. There is a large list of those type of articles. Our study aims to the opposite: height is known and DBH is to be determined.
- Figure 1: Please add some toponyms. Also, the north arrow should be better presented.
R: We consider that toponyms are not necessary. We improved compass symbol.
- Remove the extra spaces in Equation 2.
R: Fixed.
- Line 132: Please remove "comma" after a1.
R: We did not find that “comma”.
- Lines 132, 136, and 151: What is the error term?
R: Fixed.
- Line 139: than > that
R: Fixed.
- Discussion: I would remove the references to Figures and Tables in the Discussion section.
R: References to tables and figures were removed.
- References: Should be styled into the MDPI format.
R: Fixed
Reviewer 2 Report
The scientific and practical level of the paper is quite high, corresponds to the level of the journal. The text is presented clearly and logically. The level of presentation of the manuscript provides a clear understanding for the compilation of allometric equations to estimate DBH, stem volume, and whole-tree above-stump biomass for individual E. globulus, E. nitens, and P. radiata trees. The disadvantage of the described method may be its complexity, but the possibility of practical implementation using existing UAV technologies will eliminate this disadvantage.
For a more complete demonstration of all the advantages and increasing the level of reader interest in the study, the Introduction section can be supplemented with information on the practical application of these equations by a forest farmer, and at the end of the discussion section it is appropriate to add a paragraph with an answer to the question: what is planned to be investigated in the future?
The presence of the conclusion will help the reader who reads to the end to summarize the main achievements of the study.
The list of references is arranged through a very large line spacing.
I believe that the article can be published in the journal with minor changes.
Author Response
The scientific and practical level of the paper is quite high, corresponds to the level of the journal. The text is presented clearly and logically. The level of presentation of the manuscript provides a clear understanding for the compilation of allometric equations to estimate DBH, stem volume, and whole-tree above-stump biomass for individual E. globulus, E. nitens, and P. radiata trees. The disadvantage of the described method may be its complexity, but the possibility of practical implementation using existing UAV technologies will eliminate this disadvantage.
R: Thanks, we agree. The equations provided are meant for remote sensing techniques applications.
For a more complete demonstration of all the advantages and increasing the level of reader interest in the study, the Introduction section can be supplemented with information on the practical application of these equations by a forest farmer, and at the end of the discussion section it is appropriate to add a paragraph with an answer to the question: what is planned to be investigated in the future?
R: Text was included on Introduction (lines 68-80) and Discussion (lines 386-390) sections.
The list of references is arranged through a very large line spacing.
R: Fixed
I believe that the article can be published in the journal with minor changes.
R: Thanks.
Reviewer 3 Report
Using three models, the authors tried to develop equations for the three species to predict tree DBH, stem volume, and whole-tree above-stump biomass. They did a good job of achieving the research goal. I think this manuscript is suitable for publication in forests. There are some questions as follows.
(Line 158) The authors used 10-fold cross-validation to evaluate the predictive ability. It can show the objective comparison results, but the authors should mention the ratio of the training set and the validation set.
(Line 363) The authors mentioned that "the equation can be used over a wide range of sites and management conditions." Is it means the equation can be used for the whole country (Chile)? If the answer is yes, the authors should show the application results in the other test site in Chile because Chile's geographic shape is like a long stick from the North to the South. These geographic characteristics can make differences in climate by the latitude difference. Or, why don't you separate test data (20%) from original data, then make equations using separated original data (80%)? After that, test equations to the separated test data (20%). These results can support your theory more objectively.
Author Response
Using three models, the authors tried to develop equations for the three species to predict tree DBH, stem volume, and whole-tree above-stump biomass. They did a good job of achieving the research goal. I think this manuscript is suitable for publication in forests. There are some questions as follows.
(Line 158) The authors used 10-fold cross-validation to evaluate the predictive ability. It can show the objective comparison results, but the authors should mention the ratio of the training set and the validation set.
R: Our dataset is reduced, as biomass sampling is expensive and time consuming. Due to that, we decided not to split the dataset, using all sample trees for model fitting and use 10-fold cross-validation for model validation. The same approach was already used in a previous study (Biomass and Bioenergy. 155 (2021) 106280. https://doi.org/10.1016/j.biombioe.2021.106280).
(Line 363) The authors mentioned that "the equation can be used over a wide range of sites and management conditions." Is it means the equation can be used for the whole country (Chile)? If the answer is yes, the authors should show the application results in the other test site in Chile because Chile's geographic shape is like a long stick from the North to the South. These geographic characteristics can make differences in climate by the latitude difference. Or, why don't you separate test data (20%) from original data, then make equations using separated original data (80%)? After that, test equations to the separated test data (20%). These results can support your theory more objectively.
R: The equations can be used over the range of ages, stand density and tree sizes described in Table 1. Even though commercial plantations of the three species are mainly located within the geographic area described in Figure 1, there are some areas with existing stands beyond the geographic location of our sample sites. As we state at the end of discussion section, we recommend using the reported equations within the range of the dataset shown in Table 2. We evaluated the alternative you mention, but due to reduced dataset, we decided to use all data available in order to obtain the best possible equations, and using 10-fold cross validation to validate it